# MANIFOLD MICRO-SURGERY WITH LINEARLY NEARLY EUCLIDEAN METRICS

## ABSTRACT

The Ricci flow is a method of manifold surgery, which can trim manifolds to more regular. However, in most cases, the Rich flow tends to develop singularities and lead to divergence of the solution. In this paper, we propose linearly nearly Euclidean metrics to assist manifold micro-surgery, which means that we prove the dynamical stability and convergence of such metrics under the Ricci-DeTurck flow. From the information geometry and mirror descent points of view, we give the approximation of the steepest descent gradient flow on the linearly nearly Euclidean manifold with dynamical stability. In practice, the regular shrinking or expanding of Ricci solitons with linearly nearly Euclidean metrics will provide a geometric optimization method for the solution on a manifold.

## 1 INTRODUCTION

In general relativity (Wald, 2010), a complete Riemannian manifold $(\mathcal{M}, g)$ endowed with a linearly nearly flat spacetime metric $g_{ij}$ is considered for linearized gravity to solve the Newtonian limit. The form of this metric is $g_{ij} = \eta_{ij} + \gamma_{ij}$, where $\eta_{ij}$ represents a flat Minkowski metric whose background is special relativity and $\gamma_{ij}$ is small from $\eta_{ij}$. An adequate definition of "smallness" in this context is that the components of $\gamma_{ij}$ are much smaller than 1 in some global inertial coordinate system of $\eta_{ij}$. Now, let us step out of the physical world and give a similar metric $g_{ij} = \delta_{ij} + \gamma_{ij}$ in Riemannian $n$-manifold $(\mathcal{M}^n, g)$, i.e. the linearly nearly Euclidean metric, where $\delta_{ij}$ represents a flat Euclidean metric and $\gamma_{ij}$ is small from $\delta_{ij}$.

A natural problem for such a linearly nearly Euclidean metric is: how does the metric evolve over time with respect to the Ricci flow while ensuring the constant topological structure? Let us review some stability analyses of different manifolds along with the Ricci flow.

For the Riemannian $n$-dimensional manifold $(\mathcal{M}^n, g)$ that is isometric to the Euclidean $n$-dimensional space $(\mathbb{R}^n, \delta)$, Schnürer et al. (Schnürer et al., 2007) have showed the stability of Euclidean space under the Ricci flow for a small $C^0$ perturbation. Koch et al. (Koch & Lamm, 2012) have given the stability of the Euclidean space along with the Ricci flow in the $L^\infty$-Norm. Moreover, for the decay of the $L^\infty$-Norm on Euclidean space, Appleton (Appleton, 2018) has given the proof of a different method. Considering the stability of integrable and closed Ricci-flat metrics, Sesum (Sesum, 2006) has proved that the convergence rate is exponential because the spectrum of the Lichnerowicz operator is discrete. Furthermore, Deruelle et al. (Deruelle & Kröncke, 2021) have proved that an asymptotically locally Euclidean Ricci-flat metric is dynamically stable under the Ricci flow, for an $L^2 \cap L^\infty$ perturbation on non-flat and non-compact Ricci-flat manifolds.

If we embed a Riemannian $n$-dimensional manifold in the neural network, then we are training the neural network on the dynamic manifold. The most famous method for training neural networks on manifolds is the natural gradient (Amari, 1998). However, for the Riemannian manifold corresponding to the KL divergence (representing the natural gradient), its stability and convergence are still unknown (Martens, 2020). As a way of manifold evolution, Ricci flow seems to be an excellent choice to ensure that neural networks are trained on dynamic and stable manifolds (Glass et al., 2020; Jejjala et al., 2020). But the research on the relationship between the two has not yet sprouted.

In this paper, we consider a complete Riemannian $n$-dimensional manifold $(\mathcal{M}^n, g)$ endowed with linearly nearly Euclidean metrics $g(t) = \delta + \gamma(t)$. **First of all**, we prove the stability of linearly nearly Euclidean manifolds under the Ricci-DeTurck flow in the $L^2$-Norm if initial metrics are

integrable and linearly stable, i.e. has a manifold structure of finite dimension. We mean that any Ricci-DeTurck flow which starts from near $g$ exists for all time and converges to a linearly nearly Euclidean metric near $g$. Note that we use the Einstein summation convention and denote generic constants by $C$ or $C_1$.

**Furthermore**, we define and construct linearly nearly Euclidean manifolds based on information geometry and mirror descent algorithm. Based on a symmetrized convex function, we obtain the linearly nearly Euclidean divergence which is used to calculate the steepest descent gradient in linearly nearly Euclidean manifolds. **Experimentally**, when we use the approximated steepest descent gradient flow to learn several neural networks on classification tasks, we observe the evolution of its metric is consistent with the micro-surgery process under the Ricci-DeTurck flow.

## 2 RICCI FLOW

Let us introduce a partial differential equation, the Ricci flow, without explanation. The concept of the Ricci flow first published by Hamilton (Hamilton et al., 1982) on the manifold $\mathcal{M}$ of a time-dependent Riemannian metric $g(t)$ with the initial metric $g_0$:

$$\frac{\partial}{\partial t} g(t) = -2 \operatorname{Ric}(g(t))$$
$$g(0) = g_0$$
(1)

where $\operatorname{Ric}$ denotes the Ricci curvature tensor whose definition can be found in Appendix A.

The purpose of the Ricci flow is to prove Thurston's Geometrization Conjecture and Poincaré Conjecture because the Ricci flow is like a surgical scalpel, trimming irregular manifolds into regular manifolds to facilitate observation and discussion (Sheridan & Rubinstein, 2006).

In general, in order to possess good geometric and topological properties, we expect the metric to become converge and round with the help of the Ricci flow. "become round" means that the solution will not shrink to a point but converge to a constant circle. However, in most cases, we do not even know the convergence of the solution and whether the solution will develop a singularity. Next, we will discuss these issues for brevity.

### 2.1 SHORT TIME EXISTENCE

To show that there exists a unique solution for a short time, we must check if the system of the Ricci flow is strongly parabolic.

**Theorem 1** *When $u : \mathcal{M} \times [0, T) \to \mathcal{E}$ is a time-dependent section of the vector bundle $\mathcal{E}$ where $\mathcal{M}$ is some Riemannian manifold, if the system of the Ricci flow is strongly parabolic at $u_0$ then there exists a solution on some time interval $[0, T)$, and the solution is unique for as long as it exists.*

*Proof.* The proofs can be found in (Ladyzhenskaia et al., 1988). $\square$

**Definition 1** *The Ricci flow is strongly parabolic if there exists $\delta > 0$ such that for all covectors $\varphi \neq 0$ and all symmetric $h_{ij} = \frac{\partial g_{ij}(t)}{\partial t} \neq 0$, the principal symbol of $-2 \operatorname{Ric}$ satisfies*

$$[-2 \operatorname{Ric}](\varphi)(h)_{ij} h^{ij} = g^{pq} \left( \varphi_p \varphi_q h_{ij} + \varphi_i \varphi_j h_{pq} - \varphi_q \varphi_i h_{jp} - \varphi_q \varphi_j h_{ip} \right) h^{ij} > \delta \varphi_k \varphi^k h_{rs} h^{rs}.$$

Since the inequality cannot always be satisfied, the Ricci flow is not strongly parabolic, which makes us unable to prove the existence of the solution based on Theorem 1.

It is possible to understand which parts have an impact on its non-parabolic by the linearization of the Ricci curvature tensor.

**Lemma 1** *The linearization of $-2 \operatorname{Ric}$ can be rewritten as*

$$D[-2 \operatorname{Ric}](h)_{ij} = g^{pq} \nabla_p \nabla_q h_{ij} + \nabla_i V_j + \nabla_j V_i + O(h_{ij})$$
$$\text{where} \quad V_i = g^{pq} \left( \frac{1}{2} \nabla_i h_{pq} - \nabla_q h_{pi} \right).$$
(2)

*Proof.* The proofs can be found in Appendix B.1. □

The term $O(h_{ij})$ will have no contribution to the principal symbol of $-2\operatorname{Ric}$, so ignoring it will not affect our discussion of this problem. By carefully observing the above equation, one finds that the impact on the non-parabolic of the Ricci flow comes from the terms in $V$, not the term $g^{pq}\nabla_p\nabla_q h_{ij}$. The solution is followed by the DeTurck Trick (DeTurck, 1983) that has a time-dependent reparameterization of the manifold:

$$\frac{\partial}{\partial t}\bar{g}(t) = -2\operatorname{Ric}(\bar{g}(t)) - \mathcal{L}_{\frac{\partial\varphi(t)}{\partial t}}\bar{g}(t)$$
$$\bar{g}(0) = \bar{g}_0 + d, \tag{3}$$

where $d$ is a symmetric (0,2)-tensor on $\mathcal{M}$. See Appendix B.2 for details. By choosing $\frac{\partial\varphi(t)}{\partial t}$ to cancel the effort of the terms in $V$, the reparameterized Ricci flow is strongly parabolic. Thus, one can say that the Ricci-DeTurck flow has a unique solution, the pullback metric, for a short time.

## 2.2 CURVATURE EXPLOSION AT SINGULARITY

In this subsection, we will present the behavior of the Ricci flow in finite time and show that the evolution of the curvature is close to divergence. The core demonstration is followed with Theorem 4, which requires some other proof as a foreshadowing.

**Theorem 2** *Given a smooth Riemannian metric $g_0$ on a closed manifold $\mathcal{M}$, there exists a maximal time interval $[0, T)$ such that a solution $g(t)$ of the Ricci flow, with $g(0) = g_0$, exists and is smooth on $[0, T)$, and this solution is unique.*

*Proof.* The proofs can be found in (Sheridan & Rubinstein, 2006). □

**Theorem 3** *Let $\mathcal{M}$ be a closed manifold and $g(t)$ a smooth time-dependent metric on $\mathcal{M}$, defined for $t \in [0, T)$. If there exists a constant $C < \infty$ for all $x \in \mathcal{M}$ such that*

$$\int_0^T \left|\frac{\partial}{\partial t}g_x(t)\right|_{g(t)} dt \le C, \tag{4}$$

*then the metrics $g(t)$ converge uniformly as $t$ approaches $T$ to a continuous metric $g(T)$ that is uniformly equivalent to $g(0)$ and satisfies*

$$e^{-C}g_x(0) \le g_x(T) \le e^C g_x(0).$$

*Proof.* The proofs can be found in Appendix B.3. □

**Corollary 1** *Let $(\mathcal{M}, g(t))$ be a solution of the Ricci flow on a closed manifold. If $|\operatorname{Rm}|_{g(t)}$ is bounded on a finite time $[0, T)$, then $g(t)$ converges uniformly as $t$ approaches $T$ to a continuous metric $g(T)$ which is uniformly equivalent to $g(0)$.*

*Proof.* The bound on $|\operatorname{Rm}|_{g(t)}$ implies one on $|\operatorname{Ric}|_{g(t)}$. Based on Equation (1), we can extend the bound on $|\frac{\partial}{\partial t}g(t)|_{g(t)}$. Therefore, we obtain an integral of a bounded quantity over a finite interval is also bounded, by Theorem 3. □

**Theorem 4** *If $g_0$ is a smooth metric on a compact manifold $\mathcal{M}$, the Ricci flow with $g(0) = g_0$ has a unique solution $g(t)$ on a maximal time interval $t \in [0, T)$. If $T < \infty$, then*

$$\lim_{t \to T}\left(\sup_{x \in \mathcal{M}} |\operatorname{Rm}_x(t)|\right) = \infty. \tag{5}$$

*Proof.* For a contradiction, we assume that $|\operatorname{Rm}_x(t)|$ is bounded by a constant. It follows from Corollary 1 that the metrics $g(t)$ converge uniformly in the norm induced by $g(t)$ to a smooth metric $g(T)$. Based on Theorem 2, it is possible to find a solution to the Ricci flow on $t \in [0, T)$ because the smooth metric $g(T)$ is uniformly equivalent to initial metric $g(0)$.

Hence, one can extend the solution of the Ricci flow after the time point $t = T$, which is the result for continuous derivatives at $t = T$. This tell us that the time $T$ of existence of the Ricci flow has not been maximal, which contradicts our assumption. In other words, $|\operatorname{Rm}_x(t)|$ is unbounded. $\quad\square$

As approaching the singular time $T$, the Riemann curvature $|\operatorname{Rm}|_{g(t)}$ becomes no longer convergent and tends to explode.

## 3  EVOLUTION OF LINEARLY NEARLY EUCLIDEAN METRICS

Next, this paper will focus on linearly nearly Euclidean metrics, proving that them can have a good performance in terms of stability, i.e., the convergence of a Ricci-DeTurck flow $\bar{g}(t)$ to a linearly nearly Euclidean metric $\bar{g}(\infty)$. Before that, we have to construct a family $\bar{g}_0$ of linearly nearly Euclidean reference metrics such that $\frac{\partial}{\partial t}\bar{g}_0(t) = O((\bar{g}(t) - \bar{g}_0(t))^2)$. Let

$$\mathcal{F} = \left\{\bar{g}(t) \in \mathcal{M}^n \mid 2\operatorname{Ric}(\bar{g}(t)) + \mathcal{L}_{\frac{\partial \varphi(t)}{\partial t}}\bar{g}(t) = 0\right\}$$

be the set of stationary points under the Ricci-DeTurck flow. We are able to establish a manifold

$$\tilde{\mathcal{F}} = \mathcal{F} \cap \mathcal{U} \tag{6}$$

where $\mathcal{U}$ is an $L^2$-neighbourhood of integral $\bar{g}_0$.

Before starting the discussion about long time stability of linearly nearly Euclidean metrics, we need some prior knowledge about short time existence (Appendix D and Appendix E). In particular, we yield

**Lemma 2** *Let $\bar{g}(t)$ be a Ricci–DeTurck flow on a maximal time interval $t \in (0, T)$ in an $L^2$ neighbourhood of $\bar{g}_0$. We have the following estimate such that:*

$$\left\|\frac{\partial}{\partial t}d_0(t)\right\|_{L^2} \le C\left\|\nabla^{\bar{g}_0(t)}\left(d(t) - d_0\right)\right\|_{L^2}^2.$$

*Proof.* Let $\{e_1(t), e_2(t), \ldots, e_n(t)\}$ be a family of $L^2$-orthonormal bases of the kernel $\ker_{L^2}$ such that $\frac{\partial}{\partial t}e_i(t)$ depends linearly on $\frac{\partial}{\partial t}d_0(t)$. For an isomorphism orthogonal projection $\Pi :$ $T_{\bar{g}_0}\tilde{\mathcal{F}} \to \ker_{L^2}$, by the Hardy inequality (Minerbe, 2009), one has similar proofs by referring the details (Deruelle & Kröncke, 2021). $\quad\square$

**Theorem 5** *Let $(\mathcal{M}^n, \bar{g}_0)$ be a linearly nearly Euclidean $n$-manifold which is linearly stable and integrable. Furthermore, there exists a constant $\alpha_{\bar{g}_0}$ such that*

$$(\Delta d(t) + \operatorname{Rm}(\bar{g}_0) * d(t), d(t))_{L^2} \le -\alpha_{\bar{g}_0}\left\|\nabla^{\bar{g}_0}h\right\|_{L^2}^2$$

*for all $\bar{g}(t) \in \tilde{\mathcal{F}}$ which is as in (6).*

*Proof.* The similar proofs can be found in (Devyver, 2014) with some minor modifications. Due to the linear stability requirement of linearly nearly Euclidean manifolds in Definition 8, $-L_{\bar{g}_0}$ is non-negative. Then there exists a positive constant $\alpha_{\bar{g}_0}$ such that

$$\alpha_{\bar{g}_0}\left(-\Delta d(t), d(t)\right)_{L^2} \le \left(-\Delta d(t) - \operatorname{Rm}(\bar{g}_0) * d(t), d(t)\right)_{L^2}.$$

By Taylor expansion, one repeatedly uses elliptic regularity and Sobolev embedding (Pacini, 2010) to obtain the estimate. $\quad\square$

**Corollary 2** *Let $(\mathcal{M}^n, \bar{g}_0)$ be a linearly nearly Euclidean $n$-manifold which is integrable. For a Ricci–DeTurck flow $\bar{g}(t)$ on a maximal time interval $t \in [0, T]$, if it satisfies $\|\bar{g}(t) - \bar{g}_0\|_{L^\infty} < \epsilon$ where $\epsilon > 0$, then there exists a constant $C < \infty$ for $t \in [0, T]$ such that the evolution inequality satisfies*

$$\|d(t) - d_0\|_{L^2}^2 \ge C\int_0^T\left\|\nabla^{\bar{g}_0(t)}\left(d(t) - d_0\right)\right\|_{L^2}^2 \, \mathrm{d}t.$$

*Proof.* Based on Equation (16), we know

$$
\begin{aligned}
\frac{\partial}{\partial t}(d(t) - d_0) =& \Delta(d(t) - d_0) + \mathrm{Rm} * (d(t) - d_0) \\
&+ F_{\bar{g}^{-1}} * \nabla^{\bar{g}_0}(d(t) - d_0) * \nabla^{\bar{g}_0}(d(t) - d_0) \\
&+ \nabla^{\bar{g}_0}\left(G_{\Gamma(\bar{g}_0)} * (d(t) - d_0) * \nabla^{\bar{g}_0}(d(t) - d_0)\right).
\end{aligned}
$$

Followed by Lemma 2 and Theorem 5, we further obtain

$$
\begin{aligned}
\frac{\partial}{\partial t}\|d(t) - d_0\|_{L^2}^2 =& 2\left(\Delta(d(t) - d_0) + \mathrm{Rm} * (d(t) - d_0), d(t) - d_0\right)_{L^2} \\
&+ \left(F_{\bar{g}^{-1}} * \nabla^{\bar{g}_0}(d(t) - d_0) * \nabla^{\bar{g}_0}(d(t) - d_0), d(t) - d_0\right)_{L^2} \\
&+ \left(\nabla^{\bar{g}_0}\left(G_{\Gamma(\bar{g}_0)} * (d(t) - d_0) * \nabla^{\bar{g}_0}(d(t) - d_0)\right), d(t) - d_0\right)_{L^2} \\
&+ \left(d(t) - d_0, \frac{\partial}{\partial t}d_0(t)\right)_{L^2} + \int_{\mathcal{M}}(d(t) - d_0) * (d(t) - d_0) * \frac{\partial}{\partial t}d_0(t)\mathrm{d}\mu \\
\leq& -2\alpha_{\bar{g}_0}\left\|\nabla^{\bar{g}_0}(d(t) - d_0)\right\|_{L^2}^2 \\
&+ C\left\|(d(t) - d_0)\right\|_{L^\infty}\left\|\nabla^{\bar{g}_0}(d(t) - d_0)\right\|_{L^2}^2 \\
&+ \left\|\frac{\partial}{\partial t}d_0(t)\right\|_{L^2}\|d(t) - d_0\|_{L^2} \\
\leq& \left(-2\alpha_{\bar{g}_0} + C \cdot \epsilon\right)\left\|\nabla^{\bar{g}_0}(d(t) - d_0)\right\|_{L^2}^2.
\end{aligned}
$$

Let $\epsilon$ be small enough that $-2\alpha_{\bar{g}_0} + C \cdot \epsilon < 0$ holds, we can find

$$
\frac{\partial}{\partial t}\|d(t) - d_0\|_{L^2}^2 \leq -C\left\|\nabla^{\bar{g}_0}(d(t) - d_0)\right\|_{L^2}^2
$$

holds. $\qquad\square$

**Theorem 6** *Let $(\mathcal{M}^n, \bar{g}_0)$ be a linearly nearly Euclidean $n$-manifold which is linearly stable and integrable. For every $\epsilon_1 > 0$, there exists a $\epsilon_2 > 0$ satisfying: For any metric $\bar{g}(t) \in \mathcal{B}_{L^2}(\bar{g}_0, \epsilon_2)$, there is a complete Ricci–DeTurck flow $(\mathcal{M}^n, \bar{g}(t))$ starting from $\bar{g}(t)$ converging to a linearly nearly Euclidean metric $\bar{g}(\infty) \in \mathcal{B}_{L^2}(\bar{g}_0, \epsilon_1)$. Note that $\mathcal{B}_{L^2}(\bar{g}_0, \epsilon)$ is the $\epsilon$-ball with respect to the $L^2$-Norm induced by $\bar{g}_0$ and centred at $\bar{g}_0$.*

*Proof.* By Lemma 4, one can find so small $\epsilon_2 > 0$ such that $d(t) \in \mathcal{B}_{L^2}(0, \epsilon_2)$ holds. By Lemma 2 and Corollary 2, we have

$$
\begin{aligned}
\|d_0(T)\|_{L^2} &\leq C\int_1^T\left\|\frac{\partial}{\partial t}d_0(t)\right\|_{L^2}\mathrm{d}t \\
&\leq C\int_1^T\left\|\nabla^{\bar{g}_0}(d(t) - d_0(t))\right\|_{L^2}^2\mathrm{d}t \\
&\leq C\|d(1) - d_0(1)\|_{L^2}^2 \leq C\|d(1)\|_{L^2}^2 \leq C \cdot (\epsilon_2)^2.
\end{aligned}
$$

Furthermore, we obtain

$$
\|d(T) - d_0(T)\|_{L^2} \leq \|d(1) - d_0(1)\|_{L^2} \leq C \cdot \epsilon_2.
$$

By the triangle inequality, we get

$$
\|d(T)\|_{L^2} \leq C \cdot (\epsilon_2)^2 + C \cdot \epsilon_2.
$$

Followed by Corollary 4 and Lemma 2, such $T$ should be pushed further outward, because

$$
\lim_{t \to +\infty}\sup\left\|\frac{\partial}{\partial t}d_0(t)\right\|_{L^2} \leq \lim_{t \to +\infty}\sup\left\|\nabla^{\bar{g}_0}(d(t) - d_0(t))\right\|_{L^2}^2 = 0.
$$

Thus, as $t$ approaches to $+\infty$, $\bar{g}(t)$ converges to $\bar{g}(\infty) = \bar{g}_0 + d_0(\infty)$. By the Euclidean Sobolev inequality (Minerbe, 2009), $d(t) - d_0(t)$ converges to 0 as $t$ goes to $+\infty$,

$$
\lim_{t \to +\infty}\|d(t) - d_0(t)\|_{L^2} \leq \lim_{t \to +\infty}C\left\|\nabla^{\bar{g}_0}(d(t) - d_0(t))\right\|_{L^2} = 0.
$$

We now conclude a result for linearly nearly Euclidean manifolds under the Ricci-DeTurck flow, which ensures a infinite time existence. $\qquad\square$

## 4 GRADIENT FLOW WITH LINEARLY NEARLY EUCLIDEAN METRICS

Consequently, we have clarified the convergence of linearly nearly Euclidean manifolds under the Ricci-DeTurck flow. Furthermore, we will consider the solution of gradient flow with linearly nearly Euclidean metrics, which will allow us to observe the neural network behavior for back-propagation in the linearly nearly Euclidean manifold. Empirically, we introduce information geometry (Amari & Nagaoka, 2000; Amari, 2016) and mirror descent algorithm (Bubeck et al., 2015) to construct the gradient flow with the help of divergences.

### 4.1 LINEARLY NEARLY EUCLIDEAN DIVERGENCE

From the perspective of information geometry and mirror descent algorithm, the metric $\bar{g}$ can be deduced by the divergence that needs to satisfy certain criteria (Basseville, 2013). We now consider two nearby points $P$ and $Q$ in a manifold $\mathcal{M}$, where these two points are expressed in coordinates as $\boldsymbol{\xi}_P$ and $\boldsymbol{\xi}_Q$, where $\boldsymbol{\xi}$ is a column vector. Moreover, the divergence is defined as half the square of an infinitesimal distance between two sufficiently close points in Definition 2.

**Definition 2** $D[P:Q]$ *is called a divergence when it satisfies the following criteria:*

*(1) $D[P:Q] \geq 0$, (2) $D[P:Q] = 0$ when and only when $P = Q$, (3) When $P$ and $Q$ are sufficiently close, by denoting their coordinates by $\boldsymbol{\xi}_P$ and $\boldsymbol{\xi}_Q = \boldsymbol{\xi}_P + d\boldsymbol{\xi}$, the Taylor expansion of $D$ is written as*

$$D[\boldsymbol{\xi}_P : \boldsymbol{\xi}_P + d\boldsymbol{\xi}] = \frac{1}{2} \sum_{i,j} \bar{g}_{ij}(\boldsymbol{\xi}_P) d\xi_i d\xi_j + O(|d\boldsymbol{\xi}|^3),$$

*and metric $\bar{g}_{ij}$ is positive-definite, depending on $\boldsymbol{\xi}_P$.*

In order to construct a linearly nearly Euclidean metric, according to Definition 2, one can construct the divergence to obtain the expression of the metric indirectly. And the advantage is that the constructed divergence can be used to calculate the gradient flow under linearly nearly Euclidean metrics. Therefore, with the assist of Definition 3, we introduce a symmetrized convex function to construct the needed divergence:

$$\phi(\boldsymbol{\xi}) = \sum_i \frac{1}{\tau^2} \log \frac{1}{2} \left( \exp(\tau \xi_i) + \exp(-\tau \xi_i) \right) = \sum_i \frac{1}{\tau^2} \log \left( \cosh(\tau \xi_i) \right) \tag{7}$$

where $\tau$ is a constant parameter.

**Definition 3** *The Bregman divergence (Bregman, 1967) $D_B[\boldsymbol{\xi} : \boldsymbol{\xi}']$ is defined as the difference between a convex function $\phi(\boldsymbol{\xi})$ and its tangent hyperplane $z = \phi(\boldsymbol{\xi}') + (\boldsymbol{\xi} - \boldsymbol{\xi}')\nabla\phi(\boldsymbol{\xi}')$, depending on the Taylor expansion at the point $\boldsymbol{\xi}'$:*

$$D_B[\boldsymbol{\xi} : \boldsymbol{\xi}'] = \phi(\boldsymbol{\xi}) - \phi(\boldsymbol{\xi}') - (\boldsymbol{\xi} - \boldsymbol{\xi}')\nabla\phi(\boldsymbol{\xi}').$$

**Theorem 7** *For a convex function $\phi$ defined by Equation (7), the linearly nearly Euclidean divergence between two points $\boldsymbol{\xi}$ and $\boldsymbol{\xi}'$ is*

$$D_{LNE}[\boldsymbol{\xi}' : \boldsymbol{\xi}] = \sum_i \left[ \frac{1}{\tau^2} \log \frac{\cosh(\tau \xi_i')}{\cosh(\tau \xi_i)} - \frac{1}{\tau}(\xi_i' - \xi_i) \tanh(\tau \xi_i) \right] \tag{8}$$

*where the Riemannian metric is*

$$\bar{g}_{ij}(\boldsymbol{\xi}(t)) = \delta_{ij} - \left[ \tanh(\tau\boldsymbol{\xi}) \tanh(\tau\boldsymbol{\xi})^\top \right]_{ij}$$

$$= \begin{bmatrix} 1 - \tanh(\tau\xi_1(t)) \tanh(\tau\xi_1(t)) & \cdots & -\tanh(\tau\xi_1(t)) \tanh(\tau\xi_n(t)) \\ \vdots & \ddots & \vdots \\ -\tanh(\tau\xi_n(t)) \tanh(\tau\xi_1(t)) & \cdots & 1 - \tanh(\tau\xi_n(t)) \tanh(\tau\xi_n(t)) \end{bmatrix}. \tag{9}$$

*Proof.* The proofs can be found in Appendix C.1. □

Based on Theorem 7, the form of the metrics $\bar{g}(t)$ constructed by the linearly nearly Euclidean divergence is consistent with the definition of linearly nearly Euclidean metrics, as long as we adjust parameter $\tau$ to satisfy Definition 7. Moreover, we have also proven that the linearly nearly Euclidean divergence satisfies the criteria of divergence followed by Definition 2.

### 4.2 WEAK APPROXIMATION OF THE GRADIENT FLOW

By the linearly nearly Euclidean divergence, we can perform micro-surgery on the neural manifold under the gradient descent. Specifically, we dynamically consider the gradient flow toward the optimal descent direction on a manifold endowed with linearly nearly Euclidean metrics.

**Lemma 3** *The steepest descent gradient flow measured by the linearly nearly Euclidean divergence is defined as*

$$\tilde{\partial}_{\boldsymbol{\xi}} = \bar{g}^{-1}(t)\partial_{\boldsymbol{\xi}} = \left[\delta_{ij} - \tanh(\tau\boldsymbol{\xi}(t))\tanh(\tau\boldsymbol{\xi}(t))^{\top}\right]^{-1}\partial_{\boldsymbol{\xi}}. \tag{10}$$

*Proof.* The proofs can be found in Appendix C.2. □

However, Lemma 3 involves inversion, which greatly consumes computing resources. In particular, we propose two methods for approximating the gradient flow: weak approximation and strong approximation respectively.

For the weak approximation of the gradient flow, we put forward higher requirements for this metric on the basis of Definition 7, which requires that the linearly nearly Euclidean metric is also a strictly diagonally-dominant matrix based on Corollary 3.

**Corollary 3** *The weak approximation of the gradient flow measured by the linearly nearly Euclidean divergence is defined as*

$$\tilde{\partial}_{\boldsymbol{\xi}} \approx \left[\delta_{ij} + \tanh(\tau\boldsymbol{\xi}(t))\tanh(\tau\boldsymbol{\xi}(t))^{\top}\right]\partial_{\boldsymbol{\xi}} \tag{11}$$

*if the metric satisfies strictly diagonally-dominant.*

*Proof.* The proofs can be found in Appendix C.3. □

### 4.3 STRONG APPROXIMATION WITH NEURAL NETWORKS

Bypassing the requirement of weak approximation in Corollary 3, our goal is to approximate the gradient flow, $\bar{g}^{-1}(t)\partial_{\boldsymbol{\xi}}$ in Lemma 3, from the assist of multi-layer perceptron (MLP) neural network. Using the neural network, we can easily yield the strong approximation of the gradient flow because a neural network with a single hidden layer and a finite number of neurons can be used to approximate a continuous function on compact subsets (Jejjala et al., 2020), which is stated by the universal approximation theorem (Cybenko, 1989; Hornik, 1991).

As an $n \times n$ symmetric matrix, $\bar{g}(t)$ can be decomposed in terms of the combination of entries $P$ and $A$, where $P$ is the entries made up of the elements of the lower triangular matrix that contains $n(n-1)/2$ real parameters and $A$ is the entries made up of the elements of the diagonal matrix that contains $n$ real parameters. During the training in Figure 1, $\tilde{g}(t)$ can be used as strong approximation of $\bar{g}^{-1}(t)$ in the gradient flow.

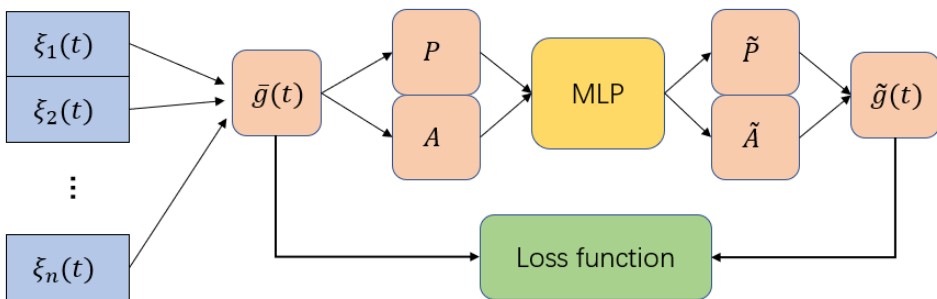

Figure 1: Flow chart of strong approximation of the gradient flow. The new entries $\tilde{P}$ and $\tilde{A}$ produced by neural network get combined into a new metric $\tilde{g}(t)$ that is used to minimize the loss function by combining with the metric $\bar{g}(t)$, where the loss function is defined by Equation (12).

$$\mathbb{L} = \|\boldsymbol{I} - \bar{g}(t)\tilde{g}(t)\|^2. \tag{12}$$

## 5   EXPERIMENT

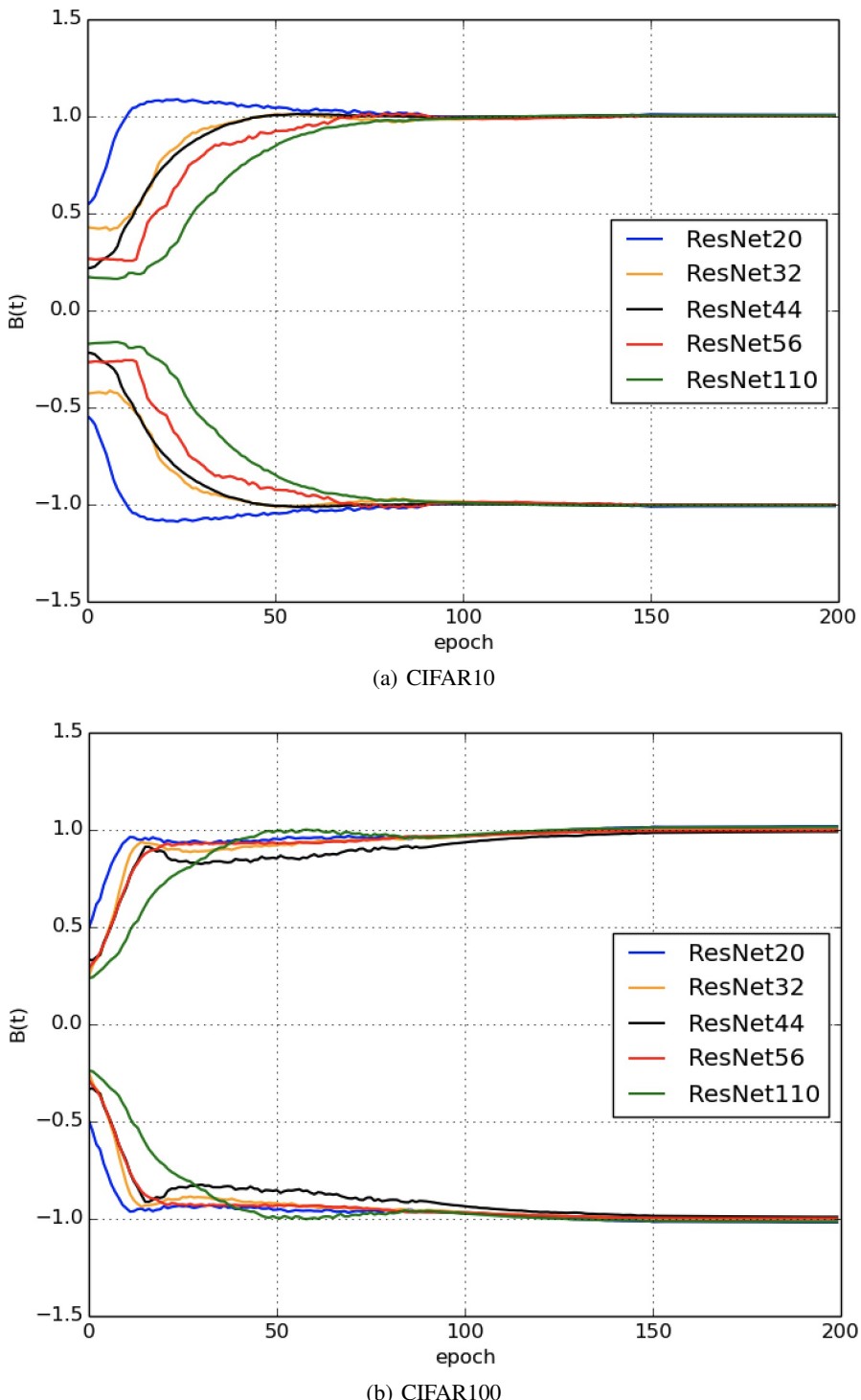

(a) CIFAR10

(b) CIFAR100

Figure 2: The evolution of metrics $\bar{g}(t)$ by the radius of a ball with the epoch of training process. Note that we use radius 1 as the calibration of a linearly nearly Euclidean metric.

**CIFAR datasets.** The two CIFAR datasets Krizhevsky et al. (2009) consist of natural color images with 32×32 pixels, respectively 50,000 training and 10,000 test images, and we hold out 5,000

training images as a validation set from the training set. CIFAR10 consists of images organized into 10 classes and CIFAR100 into 100 classes. We adopt a standard data augmentation scheme (random corner cropping and random flipping) that is widely used for these two datasets. We normalize the images using the channel means and standard deviations in preprocessing.

**Settings.** We set total training epochs as 200 where the learning strategy is lowered by 10 times at epoch 80, 150, and 190, with the initial 0.1. The learning strategy is a weight decay of 0.0001, a batch size of 128, SGD optimization. On CIFAR10 and CIFAR100 datasets, we apply ResNet20, ResNet32, ResNet44, ResNet56 and ResNet110 models (He et al., 2016) to observe the evolution of neural manifold, i.e., the convergence of metrics depended on time. As far as we define the metric $\bar{g}(t)$, we can use the length $|ds^2| = \sqrt{\sum_{i,j} \bar{g}_{ij}(t) d\xi_i d\xi_j}$ to intuitively reflect the change of metrics. Specifically, we define a ball whose radius is equal to $|ds^2|$:

$$B_r(t) := \left\{ r = \sqrt{\sum_{i,j} \bar{g}_{ij}(t) d\xi_i d\xi_j} \right\}. \tag{13}$$

**Details.** We embed the linearly nearly Euclidean manifold into a neural network, which means that a neural networks uses Corollary 3 for back-propagation. No other parts of the neural network need to be modified.

**Neural Network Behavior.** By observing the change of the ball in Figure 2, we can know the change of the metric. Through simple observation, metrics $\bar{g}(t)$ on CIFAR10 converge in about 100 epochs and metrics $\bar{g}(t)$ on CIFAR100 converge in about 150 epochs. For CIFAR10, metrics $\bar{g}(t)$ in ResNet32 and ResNet44 seem to converge the fastest. For CIFAR100, metrics $\bar{g}(t)$ in ResNet110 seem to converge the fastest. In general, experiments show that all metrics in neural manifolds converges to a linearly nearly Euclidean metric. It is consistent with the evolution of Ricci-DeTurck flow in Section 3.

**Remark.** For a neural network that is specified by connection weights, the set of all such weighs forms a manifold. When we use the gradient flow to learn a neural network ($\boldsymbol{\xi}(t)$ is composed of weights), we observe the evolution of its metric is consistent with the micro-surgery process under the Ricci-DeTurck flow. Consequently, the training of a neural manifold is also a surgery, i.e., the manifold is gradually regular, whose process is stable and eventually converges.

## 6 CONCLUSION

In this paper, we have analysed the evolution of linearly nearly Euclidean metrics under the Ricci-DeTurck flow, including proof of convergence in short and infinite time. Furthermore, we construct a linearly nearly Euclidean metric with the assist of the information geometry and use it as a springboard to reach the approximation of gradient flow. **In view of the convergence and stability of metrics, the neural network trained by the approximated gradient flow allows us to observe the relevance between Ricci flow and neural network behavior under the manifold micro-surgery**. We hope that this paper will open an exciting future direction which will use Ricci flow to assist neural network training in a manifold.

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

# A  DIFFERENTIAL GEOMETRY

1. Riemann curvature tensor (Rm) is a (1,3)-tensor defined for a 1-form $\omega$:

$$R_{ijk}^l \omega_l = \nabla_i \nabla_j \omega_k - \nabla_j \nabla_i \omega_k$$

where the covariant derivative of $F$ satisfies

$$\nabla_p F_{i_1 \ldots i_k}^{j_1 \ldots j_l} = \partial_p F_{i_1 \ldots i_k}^{j_1 \ldots j_l} + \sum_{s=1}^{l} F_{i_1 \ldots i_k}^{j_1 \ldots q \ldots j_l} \Gamma_{pq}^{j_s} - \sum_{s=1}^{k} F_{i_1 \ldots q \ldots i_k}^{j_1 \ldots j_l} \Gamma_{pi_s}^{q}.$$

In particular, coordinate form of the Riemann curvature tensor is:

$$R_{ijk}^l = \partial_i \Gamma_{jk}^l - \partial_j \Gamma_{ik}^l + \Gamma_{jk}^p \Gamma_{ip}^l - \Gamma_{ik}^p \Gamma_{jp}^l.$$

2. Christoffel symbol in terms of an ordinary derivative operator is:

$$\Gamma_{ij}^k = \frac{1}{2} g^{kl} (\partial_i g_{jl} + \partial_j g_{il} - \partial_l g_{ij}).$$

3. Ricci curvature tensor (Ric) is a (0,2)-tensor:

$$R_{ij} = R_{pij}^p.$$

4. Scalar curvature is the trace of the Ricci curvature tensor:

$$R = g^{ij} R_{ij}.$$

5. Lie derivative of $F$ in the direction $\frac{d\varphi(t)}{dt}$:

$$\mathcal{L}_{\frac{d\varphi(t)}{dt}} F = \left( \frac{d}{dt} \varphi^*(t) F \right)_{t=0}$$

where $\varphi(t) : \mathcal{M} \to \mathcal{M}$ for $t \in (-\epsilon, \epsilon)$ is a time-dependent diffeomorphism of $\mathcal{M}$ to $\mathcal{M}$.

# B  PROOF FOR THE RICCI FLOW

## B.1  PROOF FOR LEMMA 1

**Definition 4** *The linearization of the Ricci curvature tensor is given by*

$$D[\text{Ric}](h)_{ij} = -\frac{1}{2} g^{pq} (\nabla_p \nabla_q h_{ij} + \nabla_i \nabla_j h_{pq} - \nabla_q \nabla_i h_{jp} - \nabla_q \nabla_j h_{ip}).$$

*Proof.* Based on Appendix A, we have

$$\nabla_q \nabla_i h_{jp} = \nabla_i \nabla_q h_{jp} - R_{qij}^r h_{rp} - R_{qip}^r h_{jm}.$$

Combining with Definition 4, we can obtain the deformation equation because of $\nabla_k g_{ij} = 0$,

$$D[-2\text{Ric}](h)_{ij} = g^{pq} \nabla_p \nabla_q h_{ij} + \nabla_i \left( \frac{1}{2} \nabla_j h_{pq} - \nabla_q h_{jp} \right) + \nabla_j \left( \frac{1}{2} \nabla_i h_{pq} - \nabla_q h_{ip} \right) + O(h_{ij})$$

$$= g^{pq} \nabla_p \nabla_q h_{ij} + \nabla_i V_j + \nabla_j V_i + O(h_{ij}).$$

$\square$

## B.2 DESCRIPTION OF THE DETURCK TRICK

Using a time-dependent diffeomorphism $\varphi(t)$, we express the pullback metrics $\bar{g}(t)$:

$$g(t) = \varphi^*(t)\bar{g}(t)$$

is a solution of the Ricci flow. Based on the chain rule for the Lie derivative in Appendix A, we can calculate

$$
\begin{aligned}
\frac{\partial}{\partial t} g(t) &= \frac{\partial \left( \varphi^*(t)\bar{g}(t) \right)}{\partial t} \\
&= \left( \frac{\partial \left( \varphi^*(t+\tau)\bar{g}(t+\tau) \right)}{\partial \tau} \right)_{\tau=0} \\
&= \left( \varphi^*(t)\frac{\partial \bar{g}(t+\tau)}{\partial \tau} \right)_{\tau=0} + \left( \frac{\partial \left( \varphi^*(t+\tau)\bar{g}(t) \right)}{\partial \tau} \right)_{\tau=0} \\
&= \varphi^*(t)\frac{\partial}{\partial t}\bar{g}(t) + \varphi^*(t)\mathcal{L}_{\frac{\partial \varphi(t)}{\partial t}}\bar{g}(t).
\end{aligned}
$$

With the help of Equation (1), for the reparameterized metric, we have

$$\frac{\partial}{\partial t} g(t) = \varphi^*(t)\frac{\partial}{\partial t}\bar{g}(t) + \varphi^*(t)\mathcal{L}_{\frac{\partial \varphi(t)}{\partial t}}\bar{g}(t) = -2\operatorname{Ric}(\varphi^*(t)\bar{g}(t)) = -2\varphi^*(t)\operatorname{Ric}(\bar{g}(t)).$$

The diffeomorphism invariance of the Ricci curvature tensor is used in the last step. The above equation is equivalent to

$$\frac{\partial}{\partial t}\bar{g}(t) = -2\operatorname{Ric}(\bar{g}(t)) - \mathcal{L}_{\frac{\partial \varphi(t)}{\partial t}}\bar{g}(t).$$

## B.3 PROOF FOR THEOREM 3

Considering any $x \in \mathcal{M}$, $t_0 \in [0, T)$, $V \in T_x\mathcal{M}$, we have

$$
\begin{aligned}
\left| \log\left( \frac{g_x(t_0)(V, V)}{g_x(0)(V, V)} \right) \right| &= \left| \int_0^{t_0} \frac{\partial}{\partial t}\left[ \log g_x(t)(V, V) \right] dt \right| \\
&= \left| \int_0^{t_0} \frac{\frac{\partial}{\partial t}g_x(t)(V, V)}{g_x(t)(V, V)} dt \right| \\
&\leq \int_0^{t_0} \left| \frac{\partial}{\partial t}g_x(t)\left( \frac{V}{|V|_{g(t)}}, \frac{V}{|V|_{g(t)}} \right) \right| dt \\
&\leq \int_0^{t_0} \left| \frac{\partial}{\partial t}g_x(t) \right|_{g(t)} dt \\
&\leq C.
\end{aligned}
$$

By exponentiating both sides of the above inequality, we have

$$e^{-C}g_x(0)(V, V) \leq g_x(t_0)(V, V) \leq e^C g_x(0)(V, V).$$

This inequality can be rewritten as

$$e^{-C}g_x(0) \leq g_x(t_0)(V, V) \leq e^C g_x(0)(V, V)$$

because it holds for any $V$. Thus, the metrics $g(t)$ are uniformly equivalent to $g(0)$.

Now, we have the well-defined integral:

$$g_x(T) - g_x(0) = \int_0^T \frac{\partial}{\partial t}g_x(t)dt.$$

We say that this integral is well-defined because of two reasons. Firstly, as long as the metrics are smooth, the integral exists. Secondly, the integral is absolutely integrable. Based on the norm inequality induced by $g(0)$, one has

$$|g_x(T) - g_x(t)|_{g(0)} \leq \int_t^T \left| \frac{\partial}{\partial t}g_x(t) \right|_{g(0)} dt.$$

For each $x \in \mathcal{M}$, the above integral will approach to zero as $t$ approaches $T$. Because $\mathcal{M}$ is compact, the metrics $g(t)$ converge uniformly to a continuous metric $g(T)$ which is uniformly equivalent to $g(0)$ on $\mathcal{M}$. Moreover, we can show that

$$e^{-C} g_x(0) \leq g_x(T) \leq e^{C} g_x(0).$$

## C  PROOF FOR THE INFORMATION GEOMETRY

### C.1  PROOF FOR THEOREM 7

The linearly nearly Euclidean divergence can be defined between two nearby points $\boldsymbol{\xi}$ and $\boldsymbol{\xi}'$, where the first derivative of the linearly nearly Euclidean divergence w.r.t. $\boldsymbol{\xi}'$ is:

$$\partial_{\boldsymbol{\xi}'} D_{LNE}[\boldsymbol{\xi}' : \boldsymbol{\xi}]$$
$$= \sum_i \left[ \partial_{\boldsymbol{\xi}'} \frac{1}{\tau^2} \log \cosh(\tau \xi_i') - \partial_{\boldsymbol{\xi}'} \frac{1}{\tau^2} \log \cosh(\tau \xi_i) - \frac{1}{\tau} \partial_{\boldsymbol{\xi}'} (\xi_i' - \xi_i) \tanh(\tau \xi_i) \right]$$
$$= \sum_i \partial_{\boldsymbol{\xi}'} \frac{1}{\tau^2} \log \cosh(\tau \xi_i') - \frac{1}{\tau} \tanh(\tau \boldsymbol{\xi}).$$

The second derivative of the linearly nearly Euclidean divergence w.r.t. $\boldsymbol{\xi}'$ is:

$$\partial_{\boldsymbol{\xi}'}^2 D_{LNE}[\boldsymbol{\xi}' : \boldsymbol{\xi}] = \sum_i \partial_{\boldsymbol{\xi}'}^2 \frac{1}{\tau^2} \log \cosh(\tau \xi_i').$$

We deduce the Taylor expansion of the linearly nearly Euclidean divergence at $\boldsymbol{\xi}' = \boldsymbol{\xi}$:

$$D_{LNE}[\boldsymbol{\xi}' : \boldsymbol{\xi}] \approx D_{LNE}[\boldsymbol{\xi} : \boldsymbol{\xi}] + \left( \sum_i \partial_{\boldsymbol{\xi}'} \frac{1}{\tau^2} \log \cosh(\tau \xi_i') - \frac{1}{\tau} \tanh(\tau \boldsymbol{\xi}) \right)^\top \bigg|_{\boldsymbol{\xi}'=\boldsymbol{\xi}} d\boldsymbol{\xi}$$
$$+ \frac{1}{2} d\boldsymbol{\xi}^\top \left( \sum_i \partial_{\boldsymbol{\xi}'}^2 \frac{1}{\tau^2} \log \cosh(\tau \xi_i') \right) \bigg|_{\boldsymbol{\xi}'=\boldsymbol{\xi}} d\boldsymbol{\xi}$$
$$= 0 + 0 + \frac{1}{2\tau^2} d\boldsymbol{\xi}^\top \partial \left[ \frac{\partial \cosh(\tau \boldsymbol{\xi})}{\cosh(\tau \boldsymbol{\xi})} \right] d\boldsymbol{\xi}$$
$$= \frac{1}{2\tau^2} d\boldsymbol{\xi}^\top \frac{\partial^2 \cosh(\tau \boldsymbol{\xi}) \cosh(\tau \boldsymbol{\xi}) - \partial \cosh(\tau \boldsymbol{\xi}) \partial \cosh(\tau \boldsymbol{\xi})^\top}{\cosh^2(\tau \boldsymbol{\xi})} d\boldsymbol{\xi}$$
$$= \frac{1}{2\tau^2} d\boldsymbol{\xi}^\top \left( \frac{\partial^2 \cosh(\tau \boldsymbol{\xi})}{\cosh(\tau \boldsymbol{\xi})} - \tau^2 \left[ \frac{\sinh(\tau \boldsymbol{\xi})}{\cosh(\tau \boldsymbol{\xi})} \right] \left[ \frac{\sinh(\tau \boldsymbol{\xi})}{\cosh(\tau \boldsymbol{\xi})} \right]^\top \right) d\boldsymbol{\xi}$$
$$= \frac{1}{2} \sum_{i,j} \delta_{ij} - \left[ \tanh(\tau \boldsymbol{\xi}) \tanh(\tau \boldsymbol{\xi})^\top \right]_{ij} d\xi_i d\xi_j.$$

### C.2  PROOF FOR LEMMA 3

We would to know in which direction minimizes the loss function with the constraint of the linearly nearly Euclidean divergence, so that we do the minimization:

$$d\boldsymbol{\xi}^* = \underset{d\boldsymbol{\xi} \text{ s.t. } D_{LNE}[\boldsymbol{\xi}:\boldsymbol{\xi}+d\boldsymbol{\xi}]=c}{\arg\min} \mathbb{L}(\boldsymbol{\xi} + d\boldsymbol{\xi})$$

where $c$ is the constant. The loss function descends along the manifold with constant speed, regardless the curvature.

Now, we write the minimization in Lagrangian form. Combined with Theorem 7, the linearly nearly Euclidean divergence can be approximated by its second order Taylor expansion. Approximating $\mathbb{L}(\boldsymbol{\xi} + d\boldsymbol{\xi})$ with it first order Taylor expansion, we get:

$$d\boldsymbol{\xi}^* = \underset{d\boldsymbol{\xi}}{\arg\min} \mathbb{L}(\boldsymbol{\xi} + d\boldsymbol{\xi}) + \lambda \left( D_{LNE}[\boldsymbol{\xi} : \boldsymbol{\xi} + d\boldsymbol{\xi}] - c \right)$$
$$\approx \underset{d\boldsymbol{\xi}}{\arg\min} \mathbb{L}(\boldsymbol{\xi}) + \partial_{\boldsymbol{\xi}} \mathbb{L}(\boldsymbol{\xi})^\top d\boldsymbol{\xi} + \frac{\lambda}{2} d\boldsymbol{\xi}^\top \bar{g}_{ij}(t) d\boldsymbol{\xi} - c\lambda.$$

To solve this minimization, we set its derivative w.r.t. $d\boldsymbol{\xi}$ to zero:

$$
\begin{aligned}
0 &= \frac{\partial}{\partial d\boldsymbol{\xi}}\mathbb{L}(\boldsymbol{\xi}) + \partial_{\boldsymbol{\xi}}\mathbb{L}(\boldsymbol{\xi})^{\top}d\boldsymbol{\xi} + \frac{\lambda}{2}d\boldsymbol{\xi}^{\top}\left[\delta_{ij} - \tanh(\tau\boldsymbol{\xi}(t))\tanh(\tau\boldsymbol{\xi}(t))^{\top}\right]d\boldsymbol{\xi} - c\lambda \\
&= \partial_{\boldsymbol{\xi}}\mathbb{L}(\boldsymbol{\xi}) + \lambda\left[\delta_{ij} - \tanh(\tau\boldsymbol{\xi}(t))\tanh(\tau\boldsymbol{\xi}(t))^{\top}\right]d\boldsymbol{\xi} \\
d\boldsymbol{\xi} &= -\frac{1}{\lambda}\left[\delta_{ij} - \tanh(\tau\boldsymbol{\xi}(t))\tanh(\tau\boldsymbol{\xi}(t))^{\top}\right]^{-1}\partial_{\boldsymbol{\xi}}\mathbb{L}(\boldsymbol{\xi})
\end{aligned}
$$

where a constant factor $1/\lambda$ can be absorbed into learning rate. Up to now, we get the optimal descent direction, i.e., the opposite direction of gradient which takes into account the local curvature defined by $\left[\delta_{ij} - \tanh(\tau\boldsymbol{\xi}(t))\tanh(\tau\boldsymbol{\xi}(t))^{\top}\right]^{-1}$.

### C.3 Proof for Corollary 3

**Definition 5** *For $A \in \mathcal{R}^{n \times n}$, $A$ is called as the strictly diagonally-dominant matrix when it satisfies*

$$
\left|a_{ii}\right| > \sum_{j=1, j \neq i}^{n}\left|a_{ij}\right|, \quad i = 1, 2, \ldots, n.
$$

**Definition 6** *If $A \in \mathcal{R}^{n \times n}$ is a strictly diagonally-dominant matrix, then $A$ is a nonsingular matrix together.*

Subsequently, we can consider the inverse matrix of the metric $\bar{g}(t)$. Due to the strictly diagonally-dominant feature in Definition 5 and Definition 6, we can approximate $\left[\delta_{ij} - \tanh(\tau\boldsymbol{\xi}(t))\tanh(\tau\boldsymbol{\xi}(t))^{\top}\right]^{-1}$. As for we can also ignore the fourth-order small quantity $\sum O(\rho_a\rho_b\rho_c\rho_d)$ because of the characteristic of the strictly diagonally-dominant, so that

$$
\begin{aligned}
&\left[\delta_{ij} - \tanh(\tau\boldsymbol{\xi}(t))\tanh(\tau\boldsymbol{\xi}(t))^{\top}\right]\left[\delta_{ij} + \tanh(\tau\boldsymbol{\xi}(t))\tanh(\tau\boldsymbol{\xi}(t))^{\top}\right] \\
&= \begin{bmatrix} 1 - \rho_1\rho_1 & -\rho_1\rho_2 & \cdots \\ -\rho_2\rho_1 & 1 - \rho_2\rho_2 & \cdots \\ \vdots & \vdots & \ddots \end{bmatrix}\begin{bmatrix} 1 + \rho_1\rho_1 & \rho_1\rho_2 & \cdots \\ \rho_2\rho_1 & 1 + \rho_2\rho_2 & \cdots \\ \vdots & \vdots & \ddots \end{bmatrix} \\
&= \begin{bmatrix} 1 - \sum O(\rho_a\rho_b\rho_c\rho_d) & \rho_1\rho_2 - \rho_1\rho_2 - \sum O(\rho_a\rho_b\rho_c\rho_d) & \cdots \\ -\rho_2\rho_1 + \rho_2\rho_1 - \sum O(\rho_a\rho_b\rho_c\rho_d) & 1 - \sum O(\rho_a\rho_b\rho_c\rho_d) & \cdots \\ \vdots & \vdots & \ddots \end{bmatrix} \approx \boldsymbol{I}.
\end{aligned}
$$

Note that the Euclidean metric $\delta_{ij}$ is equal to the identity matrix $\boldsymbol{I}$.

## D Analysis on Linearly Nearly Euclidean Metrics

Let us give the definition of linearly nearly Euclidean metrics without further explanation:

**Definition 7** *A complete Riemannian $n$-manifold $(\mathcal{M}^n, g_0)$ is said to be linearly nearly Euclidean with one end of order $\tau > 0$ if there exists a compact set $K \subset \mathcal{M}$, a radius $r$, a point $x$ in $\mathcal{M}$ and a diffeomorphism satisfying $\phi : \mathcal{M}\backslash K \rightarrow (\mathbb{R}^n\backslash B(x, r))/SO(n)$, where $B(x, r)$ is the ball and $SO(n)$ is a finite group acting freely on $\mathbb{R}^n\backslash\{0\}$, then*

$$
\left|\partial^k(\phi_*\gamma_0)\right|_{\delta} = O(r^{-\tau-k}) \quad \forall k \geq 0 \tag{14}
$$

*holds on $(\mathbb{R}^n\backslash B(x, r))/SO(n)$. $g_0$ can be linearly decomposed into a form containing the Euclidean metric $\delta$:*

$$
g_0(t) = \delta + \gamma_0(t). \tag{15}
$$

In this paper, we consider the linear stability and integrability of the initial metric $g_0$. Fortunately, similar to the proof process of (Koiso, 1983; Besse, 2007), we can proceed that $g_0$ is integral and linearly stable.

**Definition 8** *A complete linearly nearly Euclidean $n$-manifold $(\mathcal{M}^n, g_0)$ is said to be linearly stable if the $L^2$ spectrum of the Lichnerowicz operator $L_{g_0} := \Delta_{g_0} + 2\operatorname{Rm}(g_0)*$ is in $(-\infty, 0]$ where $\Delta_{g_0}$ is the Laplacian, when $L_{g_0}$ acting on $d_{ij}$ satisfies*

$$
\begin{aligned}
L_{g_0}(d) &= \Delta_{g_0} d + 2\operatorname{Rm}(g_0) * d \\
&= \Delta_{g_0} d + 2\operatorname{Rm}(g_0)_{iklj} d_{mn} g_0^{km} g_0^{ln}.
\end{aligned}
$$

**Definition 9** *A $n$-manifold $(\mathcal{M}^n, g_0)$ is said to be integrable if a neighbourhood of $g_0$ has a smooth structure.*

# E    SHORT TIME CONVERGENCE IN THE $L^2$-NORM

For convenience, we rewrite the Ricci-DeTurck flow (3) in terms of the difference $d(t) := \bar{g}(t) - \bar{g}_0$, such that

$$
\begin{aligned}
\frac{\partial}{\partial t} d(t) = \frac{\partial}{\partial t} \bar{g}(t) &= -2\operatorname{Ric}(\bar{g}(t)) + 2\operatorname{Ric}(\bar{g}_0) + \mathcal{L}_{\frac{\partial \varphi'(t)}{\partial t}} \bar{g}_0 - \mathcal{L}_{\frac{\partial \varphi(t)}{\partial t}} \bar{g}(t) \\
&= \Delta d(t) + \operatorname{Rm}*d(t) + F_{\bar{g}^{-1}} * \nabla^{\bar{g}_0} d(t) * \nabla^{\bar{g}_0} d(t) + \nabla^{\bar{g}_0}\left(G_{\Gamma(\bar{g}_0)} * d(t) * \nabla^{\bar{g}_0} d(t)\right),
\end{aligned}
\tag{16}
$$

where the tensors $F$ and $G$ depend on $\bar{g}^{-1}$ and $\Gamma(\bar{g}_0)$. Note that $\bar{g}_0$ is a linearly nearly Euclidean metric which satisfies the above formula, where $d_0(t) = \bar{g}_0(t) - \bar{g}_0$, so that $d(t) - d_0(t) = \bar{g}(t) - \bar{g}_0(t)$ holds. Note that $\| \cdot \|_{L^2}$ or $\| \cdot \|_{L^\infty}$ denotes the $L^2$-Norm or $L^\infty$-Norm with respect to the metric $\bar{g}_0$.

**Lemma 4** *Let $(\mathcal{M}^n, \bar{g}_0)$ be a complete linearly nearly Euclidean $n$-manifold. If $\bar{g}(0)$ is a metric satisfying $\|\bar{g}(0) - \bar{g}_0\|_{L^\infty} < \epsilon$ where $\epsilon > 0$, then there exists a constant $C < \infty$ and a unique Ricci–DeTurck flow $\bar{g}(t)$ that satisfies*

$$
\|\bar{g}(t) - \bar{g}_0\|_{L^\infty} < C\|\bar{g}(0) - \bar{g}_0\|_{L^\infty} < C \cdot \epsilon.
$$

*If a Ricci-DeTurck flow in $\mathcal{B}_{L^\infty}(\bar{g}_0, \epsilon)$ for $t \geq 1$, there exist constants such that*

$$
\left\|\nabla^k\left(\bar{g}(t) - \bar{g}_0\right)\right\|_{L^\infty} < C(k)\epsilon, \quad \forall k \in \mathbb{N}.
$$

*Proof.* The similar statement for the case of negative Einstein metrics is given in (Bamler, 2010). The proofs can be translated easily to the case of linearly nearly Euclidean metrics by referring the details (Bamler, 2011). □

**Lemma 5** *Let $(\mathcal{M}^n, \bar{g}_0)$ be a linearly nearly Euclidean $n$-manifold. For a Ricci–DeTurck flow $\bar{g}(t)$ on a maximal time interval $t \in [0, T)$, if it satisfies $\|\bar{g}(0) - \bar{g}_0\|_{L^\infty} < \epsilon$ where $\epsilon > 0$, then there exists a constant $C < \infty$ for $t \in (0, T)$ such that*

$$
\|\bar{g}(t) - \bar{g}_0\|_{L^2} < C.
\tag{17}
$$

*Proof.* Based on Lemma 4, we can consider about $\|\bar{g}(t) - \bar{g}_0\|_{L^2}$. Let $\kappa$ be a function such that $\kappa = 1$ on $B(x, r)$, $\kappa = 0$ on $\mathcal{M}^n \backslash B(x, 2r)$ and $|\nabla \kappa| \leq 2/r$ where $x \in \mathcal{M}^n$ and a radius $r$.

Followed by Equation (16), we obtain

$$
\begin{aligned}
\frac{\partial}{\partial t} \int_{\mathcal{M}} |d(t)|^2 \kappa^2 \mathrm{d}\mu \leq & 2 \int_{\mathcal{M}} \left\langle \Delta d(t), \kappa^2 d(t) \right\rangle \mathrm{d}\mu + C \|\operatorname{Rm}\|_{L^\infty} \int_{\mathcal{M}} |d(t)|^2 \kappa^2 \mathrm{d}\mu \\
& + C \|d(t)\|_{L^\infty} \int_{\mathcal{M}} |\nabla d(t)|^2 \kappa^2 \mathrm{d}\mu + \int_{\mathcal{M}} \left\langle \nabla (G_\Gamma * d * \nabla d), d \right\rangle \kappa^2 \mathrm{d}\mu \\
\leq & -2 \int_{\mathcal{M}} |\nabla d(t)|^2 \kappa^2 \mathrm{d}\mu + C \int_{\mathcal{M}} |\nabla d(t)||d(t)||\nabla \kappa| \kappa \mathrm{d}\mu \\
& + C\left(\bar{g}_0\right) \int_{\mathcal{M}} |d(t)|^2 \kappa^2 \mathrm{d}\mu + C \|d(t)\|_{L^\infty} \int_{\mathcal{M}} |\nabla d(t)|^2 \kappa^2 \mathrm{d}\mu \\
\leq & (-2 + C \cdot \epsilon + C_1) \int_{\mathcal{M}} |\nabla d(t)|^2 \kappa^2 \mathrm{d}\mu + C\left(\bar{g}_0\right) \int_{\mathcal{M}} |d(t)|^2 \kappa^2 \mathrm{d}\mu \\
& + \frac{1}{C_1} \int_{\mathcal{M}} |d(t)|^2 |\nabla \kappa|^2 \mathrm{d}\mu \\
\leq & \left( C\left(\bar{g}_0\right) + \frac{2}{C_1 r^2} \right) \int_{B(x,2r)} |d(t)|^2 \mathrm{d}\mu.
\end{aligned}
$$

Note that we can always find a suitable $C_1$ to make the above formula true. By integration in time $t$, we can further obtain

$$
\int_{\mathcal{M}} |d(t)|^2 \kappa^2 \mathrm{d}\mu \leq \int_{\mathcal{M}} |d(0)|^2 \kappa^2 \mathrm{d}\mu + \left( C\left(\bar{g}_0\right) + \frac{2}{C_1 r^2} \right) \int_0^t \int_{B(x,2r)} |d(s)|^2 \mathrm{d}\mu \mathrm{d}s < \infty.
$$

Consequently, we can find a finite ball that satisfies this estimate. $\qquad\square$

**Corollary 4** *Based on Lemma 5, we further have*

$$
\sup \int_{\mathcal{M}} |d(t)|^2 \kappa^2 \mathrm{d}\mu < \infty. \tag{18}
$$

*Proof.* We obtain

$$
\begin{aligned}
\sup \int_{\mathcal{M}} |d(t)|^2 \kappa^2 \mathrm{d}\mu \leq & \sup \int_{\mathcal{M}} |d(0)|^2 \kappa^2 \mathrm{d}\mu \\
& + N \left( C\left(\bar{g}_0\right) + \frac{2}{C_1 r^2} \right) \int_0^t \sup \int_{\mathcal{M}} |d(s)|^2 \kappa^2 \mathrm{d}\mu \mathrm{d}s,
\end{aligned}
$$

where each ball of radius $2r$ on $\mathcal{M}$ can be covered by $N$ balls of radius $r$ because $(\mathcal{M}^n, \bar{g}_0)$ is linearly nearly Euclidean. By the Gronwall inequality, we have

$$
\sup \int_{\mathcal{M}} |d(t)|^2 \kappa^2 \mathrm{d}\mu \leq \exp \left( N \left( C\left(\bar{g}_0\right) + \frac{2}{C_1 r^2} \right) t \right) \sup \int_{\mathcal{M}} |d(0)|^2 \kappa^2 \mathrm{d}\mu.
$$

For the $L^2$-Norm, the Ricci-DeTurck flow in linearly nearly Euclidean manifolds has a solution for a short time. $\qquad\square$

