# OpenReview forum: "Manifold Micro-Surgery with Linearly Nearly Euclidean Metrics"
_ICLR.cc/2022/Conference — ICLR 2022 Submitted_

### Official Review · Reviewer_HjEu · 2021-11-01

**Correctness:** 3
**Technical Novelty And Significance:** 3
**Empirical Novelty And Significance:** Not applicable
**Recommendation:** 5
**Confidence:** 3

**Main Review:**

The paper presents seemingly new theoretical results and sets the stage for the analysis of gradient flow used for training neural networks. For me, the most interesting finding is the remark in Sec 4.2 stating that "When we use the gradient flow to learn a neural network ($\xi(t)$ is composed of weights), we observe the evolution of its metric is consistent with the micro-surgery process under
the Ricci-DeTurck flow". Unfortunately, this argument is not well supported in the paper, and only a small empirical test is shown in Appendix C.4.

Considering the venue (ICLR), I believe the paper is currently unbalanced. On the one hand, this is a theoretical paper presenting new analysis, which is interesting and is a contribution by itself. On the other hand, too much focus is put on the math, and I am sure how much it is of interest to the ML community. In contrast, the utility of the analysis for training neural networks using gradient flow is of major importance to the community, but it is not well treated in the paper. If the authors wish to publish their work in such ML conferences, I would recommend moving much of the analysis into the appendix and focusing instead on the utility, by describing in more detail the setting and the connection between the analysis of the evolution of this linearly nearly Eulicdean metric under the Ricci flow and the gradient flow and/or by extending the experimental study significantly.

**Summary Of The Paper:**

This paper investigates linearly nearly Euclidean metrics on Riemannian manifolds under the Ricci flow. Analysis, focusing on the stability and convergence of the evolution of such metrics, is presented. In addition, the utility of the analysis for measuring the approximation of gradient flow in the context of training neural networks is demonstrated.

**Summary Of The Review:**

Interesting new theoretical results, but the connection to gradient flow is lacking

---

> ### Author Response · Authors · 2021-11-13
> **About Experiments**
>
> We thank the reviewer for the comments and suggestion, and would like to clarify several things to address the reviewer's concerns.
>
> 1. We have added several comparison experiments to support our argument about "When we use the gradient flow to learn a neural network ( is composed of weights), we observe the evolution of its metric is consistent with the micro-surgery process under the Ricci-DeTurck flow". We embed linearly nearly Euclidean manifolds into neural networks, and observe the change of metric by the training. In general, experiments show that all metrics in neural manifolds converges to a linearly nearly Euclidean metric. In view of the convergence and stability of metrics, the neural network trained by the approximated gradient flow allows us to observe the relevance between Ricci flow and neural network behavior under the manifold micro-surgery. We hope that this paper will open an exciting future direction which will use Ricci flow to assist neural network training in a manifold.
>
> 2. We adopted the reviewer’s suggestion, moved part of the proof to the appendix, and added the experimental part. And we described the part of the gradient flow in more detail so that it can be better connected with the experiment.

---

### Official Review · Reviewer_g1ct · 2021-11-01

**Correctness:** 2
**Technical Novelty And Significance:** 2
**Empirical Novelty And Significance:** 1
**Recommendation:** 3
**Confidence:** 3

**Main Review:**

Ricci flows and the techniques involved in the study of Ricci flows such as surgery are no doubt a very exciting topic. The authors survey existing results on the Ricci flow and the occurrence of singularities. They continue to define nearly Euclidean metrics and prove various properties of these. I have not carefully checked the correctness of the survey and the presented results.

The sections on divergences, gradient flows and approximations with neural networks are largely unreadable. I did not succeed in understanding what the authors which to demonstrate here and what is the application. What do the authors mean by sentences such as "we dynamically consider the gradient flow followed with the optimal descent direction on this manifold" and "Therefore, this gradient flow is a weak approximation under the manifold micro-surgery."? What is the underlying manifold and metric in section 4.3 on approximations with neural networks?

Unfortunately, the point of the last sections is unclear to me. Because of this, it is not clear what is the contribution and point of the paper besides a survey and discussion of some very exciting mathematical topics.

**Summary Of The Paper:**

The paper concerns the Ricci flow and surgery to handle singularities occuring during the Ricci flow. The authors propose linearly nearly Euclidean metrics that they prove are stable under the Ricii-DeTurck flow. The authors use this to approximate steepest descent gradient flows in information geometry and state that they obtain a new method for geometric optimization on manifolds.

**Summary Of The Review:**

Unfortunately, while I very much appreciate the topic and the underlying mathematics, I find the paper and presentation in its present form to be too unclear that I can recommend acceptance.

---

> ### Author Response · Authors · 2021-11-13
> **About Presentation**
>
> We thank the reviewer for the comments, and would like to clarify several things to address the reviewer's concerns.
>
> 1. We reorganized the content about gradient flow. Actually, we use the linearly nearly Euclidean divergence defined in Section 4 to obtain the optimal descent direction of the neural network in the linearly nearly Euclidean manifold, which will allow us to observe the gradient descent performance (Eq.10) of the neural network on this dynamic manifold.For the same linearly nearly Euclidean manifold, both the neural network and the Ricci flow make the metric with pertubation converge to a linearly nearly Euclidean metric. In Section 4.3, the manifold is the linearly nearly Euclidean manifold, and the metric is the linearly nearly Euclidean metric. Actually, such settings are applicable to the full paper.
>
> 2. We described the part of the gradient flow and experiments in more detail so that we show the relevance between the Ritchie flow and the neural network. For back-propagation, we construct the linearly nearly Euclidean divergence to obtain the gradient flow with linearly nearly Euclidean metric.

---

### Official Review · Reviewer_ibvS · 2021-11-02

**Correctness:** 3
**Technical Novelty And Significance:** 3
**Empirical Novelty And Significance:** Not applicable
**Recommendation:** 3
**Confidence:** 2

**Main Review:**

The paper is heavily theoretic and expects a lot of background knowledge in differential geometry from the reader. I do have some background but I was not able to follow every detail in the derivations given the short review period. I am especially not familiar with the term micro-surgery for Ricci flows, there is no citation for this concept and any explanation I was able to find did not make sense for how it is used in this paper for me. The main idea of linearizing the Ricci flow to increase the robustness sounds meaningful to me, and as far as I can tell the math behind it looks convincing (though I did not check all details).

The paper does not describe the context in which it exists (as in there is no related work section, and what related work is mentioned is about properties of manifelds). It is not clear to me what the method has to do with the topics of ICLR, there is no mention of which papers use Ricci flow for anything, and the results are also not clear in how this method is used on CIFAR exactly. This is my major critisism of this paper, as someone who is not an expert in this topic, it was impossible for me to find a starting point to judge the contributions and novelty.

The main paper does not present any experiments, and the experiments in the appendix on CIFAR are neither properly explained, nor was I able to interpret them (also due to the lack of description of the experiment setup). The authors only claim that "Actually, the change in metric is not so obvious" which I must interpret as it did not really work. I see that this is a very basic theoretical work, and I would have been fine with experiments on artifical and simplistic data but the complete lack of any positive results does not raise faith.

**Summary Of The Paper:**

The paper proposes a linerization method for ricci flow that is used for manifold surgery which makes the process more robust.

**Summary Of The Review:**

First of all, this is not my area of expertise, and the review period was too short to go into all the mathematical details. Therefore, it might as well be possible that this is quite genius and I was just not able to see it. However, the paper does not explain the impact of its contribution or its applications well, does not have a proper related work section, and the results are half hidden in the appendix and their positive aspects not explained.
I have rated reject because putting contributions into context through the related work (while often annoying and seemingly just taking away space) is an essential part of a paper and skipping over this leaves the paper incomprehensible for many people.
My ratings on correctness and novelty are only approximations, and I would not oppose acceptance if the theory has strong supporters in the other reviewers.

---

> ### Author Response · Authors · 2021-11-13
> **About Experiments**
>
> We thank the reviewer for the comments, and would like to clarify several things to address the reviewer's concerns.
>
> 1. Well, we understand that the reviewer seems to be confused about the application of this paper. We have added several comparison experiments to describe the behavior of neural networks on CIFAR. We embed linearly nearly Euclidean manifolds into neural networks, and observe the change of metric by the training. In general, experiments show that all metrics in neural manifolds converges to a linearly nearly Euclidean metric. In view of the convergence and stability of metrics, the neural network trained by the approximated gradient flow allows us to observe the relevance between Ricci flow and neural network behavior under the manifold micro-surgery. We hope that this paper will open an exciting future direction which will use Ricci flow to assist neural network training in a manifold.
>
> 2. We have added details and settings about the experiment to ensure that the results of the experiment are repeatable and effective.
>
> 3. As for the application of Ricci flow, there are really no other papers besides the application of mathematical theory. Therefore, we hope that this paper can provide a beginning for the application of Ricci flow.

---

> > ### Comment · Reviewer_ibvS · 2021-11-18
> > **Reply**
> >
> > 1. I assume the comparisons are in Figure 2? As far as I understand these are all convergence results of your method, where are the comparison? How does a network without your method converge? "In general, experiments show that all metrics in neural manifolds converges to a linearly nearly Euclidean metric." Does that mean for all training methods, but yours only converges faster?
> > I am missing an intuition about what a "metric in a neural manifold" is, and as a result it is also not clear to me why having a Euclidean there is a desirable property. Especially with the lack of comparison of common training methods, the advantages are not obvious. I said in my review that the setup and placement of the experiments made it strongly seem that the method does not work well. This was not disputed here, and while the experiments are more prominent now, there is not a single indictator in the text and figures that it works better than normal training methods.
> >
> > It is roughly clear to me why B(t) = 1.0 is the optimal value, but a short explanation after Eq.(13) would help.
> > 2. The experiment setup is now much clearer.
> > 3. I think in this case especially it would be very important to make the relationship between neural network training and the Ricci flow clearer in the introduction. (and then neural networks are never mentioned again until Section 4.2) What exactly is the problem tackled here? What is the current challenge with neural network training that is improved by using this theory? These things are mentioned in the last two sentences of the introduction but there is not connection to the rest of the introduction. It is now clearer to me after carefully reading the paper multiple times but the introduction should be understandable the first time. I think the introduction is too technical, and actually more a background section. The introduction should make clear what is the problem, why is it important and explain the proposed solution in a brief and understandable way.

---

> > > ### Author Response · Authors · 2021-11-18
> > > **Identify the Problem**
> > >
> > > 1. Regarding common methods, I understand it as a training method in Euclidean space. The metric at this time has no convergence at all （keep constant）, because the Euclidean metric is an identity matrix, which is represented as a horizontal straight line in the Figure 2. We are discussing convergence behavior on a dynamic manifold, and we don’t understand why we need to compare with the metric (a straight line) in a flat space.
> > >
> > > 2. Otherwise, we have made some supplements to the introduction, and hope to help you understand this article although the research on neural networks and Ricci flow is rare:
> > >
> > > If we embed a Riemannian $n$-dimensional manifold in the neural network, then we are training the neural network on the dynamic manifold. The most famous method for training neural networks on manifolds is the natural gradient [a]. However, for the Riemannian manifold corresponding to the KL divergence (representing the natural gradient), its stability and convergence are still unknown [b]. As a way of manifold evolution, Ricci flow seems to be an excellent choice to ensure that neural networks are trained on dynamic and stable manifolds [c,d]. But the research on the relationship between the two has not yet sprouted.
> > >
> > > [a] Amari, Shun-Ichi. "Natural gradient works efficiently in learning." Neural computation 10.2 (1998): 251-276.
> > >
> > > [b] Martens, James. "New Insights and Perspectives on the Natural Gradient Method." Journal of Machine Learning Research 21 (2020): 1-76.
> > >
> > > [c] Glass, Samuel, Simeon Spasov, and Pietro Liò. "RicciNets: Curvature-guided Pruning of High-performance Neural Networks Using Ricci Flow." arXiv preprint arXiv:2007.04216 (2020).
> > >
> > > [d] Jejjala, Vishnu, Damian Kaloni Mayorga Pena, and Challenger Mishra. "Neural network approximations for Calabi-Yau metrics." arXiv preprint arXiv:2012.15821 (2020).

---

> > > > ### Comment · Reviewer_ibvS · 2021-11-29
> > > > **Still unclear**
> > > >
> > > > Apart from not understanding the theory, which I would give less weight to since I am far from an expert here, I still do not see how any of the shown experiments show any advantage this method might give.
> > > >
> > > > The experiments show how the metric converges towards Euclidean.
> > > > -> This means Euclidean metric is a preferable property.
> > > > but (if I understood correctly): gradient descend is constant and always Euclidean.
> > > > If Euclidean metric was the desirable property here, gradient descent would win.
> > > > Therefore, this experiment does not show the advantage of the proposed method.
> > > > There is also no other experiment that shows a case where training with this  method does something better than gradient descent. And I said in my initial review that I assume this means that the method does not even work well on toy examples, and this was not even disputed.
> > > >
> > > > Since I did not understand the theoretical part, I would be fine with my opinion being disregarded here. However, the other reviewers have similar concerns about the clearness of the manuscript.

---

> > > > > ### Author Response · Authors · 2021-11-30
> > > > > **Focus deviation**
> > > > >
> > > > > Well, the focus of the paper is never to compare with the Euclidean space, and the metric is not to converge to Euclidean metric, but to linearly nearly Euclidean metric. We only found the convergence of the metric for a neural network in the linearly nearly Euclidean space consistent with the behavior of Ricci flow! This paper is not to compare with the pros and cons of the Euclidean space. Naturally, the proposed method only serves to demonstrate the behavior of the neural network in back-propagation.

---

### Decision · Program_Chairs · 2022-01-20

**Decision:**

Reject

**Comment:**

Ricci flow is a central topic in geometric analysis. It has had stunning applications in mathematics, most notably the proof of the Poincare conjecture. The major issue is that it, while it can be used to make a manifold more well-behaved, it frequently develops singularities. The main contribution fo this paper is in introducing linearly nearly Euclidean metrics. They give a proof of convergence in both short and infinite time, under the Ricci-DeTurk flow, and exploit connections to information geometric and mirror descent to develop methods for approximating the gradient flow. The paper is confusingly written (compounded by poor organizational structure and many grammatical mistakes). Perhaps the biggest issue is that it does not have any clear relevance to machine learning. Some sections mention connections to neural networks, but the reviewers found these sections to be indecipherable.